# Effectiveness of Virtual and Augmented Reality for Emergency Healthcare Training: A Randomized Controlled Trial

**DOI:** 10.3390/healthcare13091034

**Published:** 2025-04-30

**Authors:** Jose Manuel Castillo-Rodríguez, Jose Luis Gómez-Urquiza, Sofía García-Oliva, Nora Suleiman-Martos

**Affiliations:** 1Granada Traumatology Hospital, Andalusian Health Service, 18013 Granada, Spain; jose_castillo130@hotmail.com; 2Nursing Department, Ceuta Faculty of Health Sciences, University of Granada, 51001 Ceuta, Spain; sofiagarcia@ugr.es; 3Nursing Department, Faculty of Health Sciences, University of Granada, 18071 Granada, Spain; norasm@ugr.es

**Keywords:** augmented reality, virtual reality, emergencies, professional training

## Abstract

**Background**: Appropriate clinical training for emergencies is a key factor in the quality of healthcare. Advances in technology facilitate the creation of new forms of training, fostering student interaction and engagement. In this respect, augmented and virtual reality approaches in healthcare training are generating great interest. **Objectives**: to evaluate the effectiveness of virtual and augmented reality in emergency healthcare training. **Methods**: A randomized controlled trial was conducted with two intervention groups. Intervention group No. 1 (*n* = 30) received in-class instruction followed by practice with a cardiopulmonary resuscitation (CPR) manikin and using virtual reality with the Meta Quest model (using the VR applications Heart, Lung, and School of CPR). Intervention group No. 2 (*n* = 31) received the same in-class training but followed by an augmented reality session. In this intervention, the Heart and Lung AR applications were used on an iPhone 15 ProMax. The control group (*n* = 32) only received in-class instruction and practice with the CPR manikin. **Results**: The virtual reality and augmented reality interventions improved learning effectiveness and user satisfaction. These results were statistically significant (*p* < 0.05) for knowledge post-intervention between groups, VR being the highest. However, the results were not significant for satisfaction. **Conclusions**: VR procedures can be effective for increasing learning effectiveness in emergency training.

## 1. Introduction

Although high-quality clinical training for the complex world of emergency attention is a crucial aspect of healthcare education, it is often under-represented in study plans, and student bodies have made urgent calls for this question to be addressed [1].

The innovative use of technology in professional training can enhance interaction and commitment in the learning environment [1,2]. Appropriately applied, technological resources can provide a safe learning environment in which the basic skills of exploration, execution and commitment can be developed and perfected, thus reducing passive exposure to training content (reading and listening) [2,3].

Technological research has highlighted the potential of this approach, and numerous training tools have been created for theoretical and practical environments, like the operating room for surgeons, in which multiple clinical situations can be addressed [4]. Historically, technology-based teaching was first used in cardiopulmonary resuscitation (CPR) training with a manikin providing interactive responses, in a realistic simulation environment for medical residents in which information from the camera and the manikin generated a real-time assessment of the participant’s performance [5,6]. This teaching method used two different devices: collaborative tools aimed at saving lives by reducing the duration of healthcare; and virtual simulation collaborative tools, which collect real-time information on the participant’s performance during clinical practice [5,7].

Augmented reality (AR) instruments integrate realistic three-dimensional virtual elements, in environment and time, that provoke a cognitive, emotional, behavioral and physiological response from the student [3,8]. This technology is based on the combination of an optical component (semi-transparent lenses that provide optical transparency to combine the elements of the virtual environment with those of the real one), a visual component (cameras that show the real world augmented with virtual elements) and a projection component (by which virtual elements are reflected in the real world without the collaboration of portable technological devices) [5,8]. This approach seeks to increase the information obtained from the practice scenario via haptic feedback and realistic interactions, and to enhance the subjective impression of participating in a realistic experience, thus improving the clinical training provided [5].

In recent years, there has been increasing research interest in optical transparent technologies, including products such as Microsoft HoloLens and Google Glass for healthcare personalized (nurses, physicians, technicians, etc.) for CPR training [9,10,11]. In healthcare training, these technologies provide a means whereby the instructor’s training experience is physically or remotely [12] transcended to medical students or to surgeons’ gastrectomy training via specific communication devices, often in the form of head-mounted instruments (which, compared to traditional training, improve accessibility, student interest and didactic value) [13,14,15].

Virtual reality (VR) instruments are used to generate scenarios that are unreal, but which immerse the interlocutor in a real-time environment very close to reality, thus provoking a cognitive, emotional, behavioral and physiological response [3]. The technology is based on projecting images and sounds that are synchronized with the user’s movements in the virtual environment, using facial screens, stereo headphones and tracking systems (high-resolution mobile devices with a high refresh rate) [2,16]. The main difference between AR and VR is that AR involves digital elements layered over the reality the person is in, while in VR the person is in a completely computer-generated environment.

The Samsung Gear VR face display has a touch interface with a grid that enables the user to interact with the virtual tools, elements or texts displayed by the Oculus application [17]. The tools have accelerometers (infrared tracking systems) that coordinate the user’s head movements in the virtual world, in each plane of execution [17]. Together, these instruments make it possible to project virtual scenarios via a 360° visual field onto a high-resolution screen, which can also be linked to smartphones [17].

Digital tools are highly versatile; among other applications, they facilitate learning and training activities for healthcare professionals by providing virtual therapeutic, diagnostic and communication environments [13,14,18]. These tools constitute an accessible, empirical teaching resource, based on interactions with virtual scenarios, that ensure patient safety, while allowing the participant to repeat the educational material (therapeutic techniques, critical situations, etc.) indefinitely [19,20]. Furthermore, these technological tools have economic benefits. Despite the currently high cost of VR technology, prices are falling [21], and there are even free applications [22] that are highly appropriate for clinical training, allowing immersive technologies to be used cost-effectively and remotely. In short, AR and VR tools for professional training may increase the achievement of results compared to those of traditional learning approaches [5,11].

The aim of the present study is to evaluate the effectiveness of VR and AR in learning outcomes (knowledge) obtained versus traditional learning in emergency and urgent healthcare training for nursing students and nurses. The secondary aim is to analyze the students’ and nurses’ satisfaction and opinions about the use and functionality of VR and AR. This study is motivated by the desire to add evidence to two technological training/teaching methods that are being used in healthcare training but have rarely been compared.

## 2. Materials and Methods

A randomized controlled trial with two intervention groups was conducted. The hypothesis of the study was that VR will be more effective than AR for emergency healthcare training and that both will be more effective than usual learning.

### 2.1. Participants

The participants in this study were nurses working in emergency units at third-level hospitals in southern Spain and nursing students from the University of Granada who were completing their internships in Emergency Room units. Nurses from other services, such as primary care, and those who worked on standby or performed sporadic 24 h shifts in emergency units, were excluded from the study.

### 2.2. Sample Size

For this study, n = 105 people were invited to participate, allowing for a loss rate of 15%. A total of 35 people were recruited for each of the three study groups. The final sample was n = 93 (88.57% participation rate). Invitations to participate in the study were sent by e-mail, with a Google Form, after consulting with the heads of the treatment units concerned.

### 2.3. Randomization

An online randomization source (https://www.ugr.es/~jsalinas/Aleatorios.htm accessed on 28 June 2024) was used to generate 105 random numbers, representing the total number of participants. These numbers were then assigned to the participants following the sequence Control Group, Intervention Group No. 1 and Intervention Group No. 2, repeating the selection until all three groups were complete. The sequencing and assignment were carried out by a single member of the research group.

### 2.4. Intervention

The intervention consisted of a one-day, 6 h training course (3 h, a 30 min break and another 3 h) related to urgent pathologies (acute myocardial infarction, acute asthmatic reaction and cardiopulmonary resuscitation, CPR). The training, in each of the three study groups, was given by the same instructor from the research team.

The control group received training according to the traditional in-class method, followed by practice with a CPR manikin.

Intervention group No. 1 (VR group) received both the same training as the control group and also the use of Meta Quest/Oculus VR glasses to view the development of acute myocardial infarction (using Heart VR application), that of acute asthmatic reaction (using Lung VR application) and the response to cardiorespiratory arrest (using the School of CPR application).

Intervention group No. 2 (AR group) received the same training as the VR group, except that the AR versions of the Heart and Lung applications were implemented on an iPhone 15 Pro Max. The applications were the same as group No. 1 but using the AR version instead.

### 2.5. Study Variables and Data Collection

Sociodemographic data (age, sex, professional level and marital status) were collected from the study participants. An ad hoc questionnaire (with 40 items and two open questions) was conducted regarding the previous use of VR and AR technologies, their experience with its use and its influence on variables related to learning (To what extent do you agree with the following statements? (1 = Not at all; 2 = Not really; 3 = Somewhat; 4 = Quite strongly; 5 = Completely)) and, finally, two open questions about the positive and the negative aspect of using RV and RA in healthcare training. The independent variable considered was the use of VR or AR in training, and the dependent variables were the participants’ level of post-training knowledge and their satisfaction with the training, which was evaluated with an ad hoc questionnaire elaborated by the lecturer. The knowledge questionnaire was administered online the day before the course began, and post-intervention data were obtained via a Google Form distributed at the end of the training before leaving the training. The knowledge questionnaire was based on 10 questions with 4 answer options about the pathologies from the training (1 point for each correct answer). Satisfaction was assessed with a 5-point Likert scale with the question: How satisfied are you with the training you received? 1: Very dissatisfied. 2: Dissatisfied. 3: Neutral. 4: Satisfied. 5: Very satisfied.

### 2.6. Blinding

Due to the characteristics of the intervention, it was only possible to blind the person who analyzed the results (evaluator blinding). Neither the participants nor the person implementing the intervention could be blinded.

### 2.7. Statistical and Qualitative Data Analysis

All statistical analyses were performed using SPSS 28 software. First, a descriptive analysis was performed with measures of location and central tendency (means, standard deviations, minimum value and maximum value) for the quantitative variables, and a frequency analysis for the categorical variables.

Secondly, the normality of the variables was verified using the Kolmogorov–Smirnov test. Inter-group post-intervention knowledge and satisfaction means were compared using an ANOVA test. Intragroup differences were assessed with Student’s t test for paired samples.

A descriptive analysis of the two open questions was performed, first by coding the response of each participant (P01, P02, P03, etc.) and subsequently identifying the main categories of the responses using ATLAS.ti 23 software.

## 3. Results

The main characteristics of the study participants are shown in Table 1. In the total sample, there were 93 initial participants, n = 32 for control group, n = 30 for VR group and n = 31 for AR group. The participants were predominantly female (75.16% women vs. 24.73% men) and the average age was 25.39 years (range: 20–31 years). The educational level of the total sample was stratified into three categories: undergraduate students (32.25% of participants), graduate and Master’s Degree (66.60%), and PhD (1.07%). Prior to the study, AR and VR were used by 1.16% and 12.91% of participants, respectively.

### 3.1. Effectiveness of AR and VR in Emergency and Urgent Healthcare Training and Student Satisfaction

Before starting AR and VR training for emergency and urgent healthcare situations, there were no statistically significant differences between the three groups in their level of knowledge—ANOVA F = 0.610 (*p* = 0.546)—as reflected in the following responses to the initial questionnaire: mean score for the AR group, 4.16 points (minimum 2 points and maximum 7 points); VR group, 4.20 points (minimum 2 points and maximum 8 points); and control group, 3.94 points (minimum 2 points and maximum 7 points).

After completing AR and VR training, the following mean scores were obtained for the final questionnaire: AR group, 8.19 points (minimum 6 points and maximum 10); VR group, 8.66 points (minimum 6 points and maximum 10); and control group, 7.98 points (minimum 6 points and maximum 10) (Table 2). The mean difference pre-post for the control group was 4.04 (95%CI 4.80, 3.44 *p* < 0.001), for the VR group, 4.46 (95%CI 5.12, 3.80 *p* < 0.001), and for the AR group, 4.03 (95%CI 4.53, 3.52 *p* < 0.001).

After completing AR and VR training for emergency and urgent healthcare, there were not statistically significant differences in student satisfaction between groups. The following mean levels of student satisfaction were recorded: AR group, 4.22 points (range 3–5 points); VR group, 4.53 points (range 3–5 points); and control group, 4.02 points (range 2–5 points). The differences in the knowledge after the intervention were statistically significant (Table 2).

Table 3 presents the mean satisfaction scores recorded by the participants regarding the use of AR and VR technologies for emergency and urgent healthcare instruction, with the following opinions. The members of the AR group awarded the highest scores (i.e., expressed the most favorable opinions) to the statements that the use of AR complemented the training, and that they were satisfied with this experience. The VR group highlighted the views that they felt physically and mentally safe during the experience and that this option should be more present in healthcare training. The views that received least support were, for the AR group, that the experience provoked anxiety and that it hindered their training and, for the VR group, that they did not like the use of VR in training and that it hindered their learning. The control group was excluded from this phase of the study, as these participants received only traditional training methods, and not AR or VR.

### 3.2. Qualitative Results About Positive and Negative Aspects of Augmented Reality

The participants reflected on their experience of using AR in emergency and urgent healthcare training. Our analysis highlights three key pillars of this experience that were identified: motivation, value and ease of use.

On several occasions, the participants expressed their appreciation of the motivation provided, for example, as follows: “*Using augmented reality for the first time was eye-opening for me…. I think it definitely has a future in teaching because seeing what you’ve just learned in the real world is very impressive [P08]. These types of approach to teaching increase our interest and motivation, very good idea* [*P12*]. *It was my first time and I found it very interesting, I haven’t got anything negative to say about it* [*P17*]”.

Regarding the value and use of this technology, the participants made the following observations: “*I think it facilitates learning [P03]. It*’*s good for learning [P19]. It was my first time and I found it educational [P23]. The experience of using AR for the first time as a learning aid was really cool, because I found it quite useful and innovative [P25]. Being able to analyse pathologies in an almost real way as if you had the heart in front of you, and to be able to modify it however you like, that’s very good for learning and you’re less likely to forget* [*P06*]*. It helps you remember the concepts you’ve learned* [*P08*]”.

However, the AR training experience was not entirely satisfactory. The participants also referred to certain drawbacks such as the modification of reality: “*Not seeing the heart or the lungs inside a body but floating within the environment, this took away from the sense of realism*” [*P18*])*,* disappointment (“*The mobile application was not as impressive as the glasses may be*” [*P15*]) or the scant educational use made of the technology (“*They weren’t used in most courses… They’ve never been used* [*P11*]”.

### 3.3. Qualitative Results About Positive and Negative Aspects of Virtual Reality

The participants also described their experience of using VR in emergency and urgent healthcare training, citing five basic characteristics: motivation, usefulness, ease of use, innovation and immersion.

The motivation provided by VR learning was referred to in the following terms (“*I really liked the experience even though I hadn*’*t tried it before* [*P01*]*. I loved it. Being able to modify the type and severity of pathologies while you’re in VR is an incredible way to learn* [*P05*]*. The educational use of the glasses seemed excellent to me even though I’m not really keen on technology* [*P09*]*. I enjoyed it, it was very good and I’m glad to have experienced it* [*P27*]*. It was fantastic to see the teaching approach and how you* “*live*” *knowledge within the virtual world, even though I had previously used VR to play, I can’t see anything wrong in it* [*P21*]”).

The participants made these comments, among others, regarding the usefulness of VR learning: “*I think it helps you understand and solidify the questions examined, to retain them in the future* [*P01*]*. It’s very good for learning* [*P04*]*. I think it’s a very good technology as a complement to the usual training* [*P13*]*. Seeing everything as if you were inside the heart or the lungs, for each of the different diseases, is useful to understand the course* [*P19*]*. It’s useful, complementing the training, I think it’s the key to improving our training* [*P20*]*. In my opinion, its usefulness in teaching is quite clear, because it allows you to* “*be present*” *within whatever*’*s happening to the patient and because it allows you to see everything directly*” [*P22*].

With respect to ease of use, the participants made these statements: “*You don’t need to be an expert to get to grips with it* [*P06*]*. It’s easy to learn how to use* [*P13*]*. Everything is good, I found it easy and intuitive to use*” [*P15*].

On the innovation associated with AR/VR, they said the following: “*It seemed very innovative, allowing me to immerse myself in what was seen in the course [P22]. Innovative way to complement the training* [*P24*]. *I think that in the next few years it will be in common use, I believe this technology has a great future in teaching*” [*P23*]).

Finally, on the immersion achieved with the use of VR technology, the members of the study sample made these comments: “*VR provided greater visual, audio and manual immersion in the questions considered; you felt you were actually inside the body, seeing what was happening and able to modify it as you chose* [*P15*]”.

As with the AR group, the participants in the VR training program also highlighted certain limitations of the experience. Firstly, the ease of use, namely that “*Handling was a bit difficult as we had never used VR before* [*P02*]*. It wasn’t easy to use, it was necessary to learn how to handle the device before using it*” [*P07*]. The participants also mentioned some aspects that produced disappointment: “*We thought the glasses would have greater image resolution* [*P17*]*. The glasses are very expensive*” and referred to discomfort: “*There should be more glasses available so that all of us could take the course at the same time without having to wait* [*P14*]*. The VR made some people a bit dizzy if they were not used to it [P02]. Sharing the glasses among so many participants was a bit off-putting because it was hot and our faces were sweaty [P08] The workshop would have gone faster if there had been more glasses, the glasses were a bit uncomfortable*” [*P08*].

## 4. Discussion

Analysis of the research findings showed that VR-based learning produces significantly more benefit to emergency and urgent healthcare training than either AR-based learning or traditional methods. Although AR achieved better results than traditional methods in this context, the difference was not statistically significant. With respect to user satisfaction, the VR approach was again preferred to AR and traditional methods. And as before, although AR learning generated greater satisfaction than traditional methods alone, the difference was not significant.

The study participants highlighted certain positive aspects of the AR experience—related to motivation, usefulness and ease of use—but also some negative considerations, such as the perceived modification of reality, disappointment with the practical implementation of the technology, some physical discomfort and inadequate didactic application. Similarly, the VR users appreciated the enhanced motivation, usefulness, ease of use, innovation and immersion achieved, but also pointed out negative aspects, especially disappointment and discomfort.

In a previous study, AR has been tested in a randomized clinical trial [23] as a teaching tool for nursing students specializing in critical care. The results showed that AR learning motivated students to a significantly greater degree than traditional training methods and thus created a more favorable learning experience (indeed, 98% of participants stated they were satisfied with this teaching method).

In another study, with randomized design characteristics [24], AR was used for medical skills acquisition in ultrasound-guided interventions during central venous catheter (CVC) placement. This study concluded that using the Foresee-X intelligent ultrasound simulation system improved cognitive load and performance scores in ultrasound-guided interventions, compared to the results achieved by a control group, and therefore that smart ultrasound simulators can be considered a useful means of improving the transfer of clinical skills in this field. However, according to another randomized study [25], despite the high immersion capacity offered by AR, the students who used this technology obtained lower scores for the acquisition of new knowledge in neuroanatomy, compared to a control group, possibly due to the “modification of reality” cited by participants in the present study. This finding puts in doubt the value of AR for educational purposes, suggesting it should be restricted to recreational applications. A third randomized trial [26] on the impact of AR on the acquisition of vascular access catheterization skills using ultrasound techniques, reported a significant shortening of the time required to perform the clinical intervention and suggested that training with AR could be a valuable complementary teaching approach for healthcare practitioners in the field of urgent and emergency healthcare. Among other benefits of the technology, the participants in this study emphasized its ease of use and the fact that the device’s transducer could be visually updated without prejudice to its functionality. These observations support the conclusions of the current research.

VR-based learning has also been considered in earlier research. The authors of a randomized trial described VR as a suitable training tool for students receiving specialized training in spinal interventions and reported that VR learning had a significantly greater impact on knowledge acquisition and on time management during the clinical intervention than was provided by traditional methods [27]. Moreover, the participants who received VR training reported a higher level of satisfaction than those in the control group. Another randomized study, on nasogastric tube feeding [28], corroborated the above finding, referring specifically to the significant perceived improvement in knowledge acquisition after using VR during clinical training. These benefits might be related to the degree of immersion provided by VR, which generates a highly realistic environment that prompts the user’s full attention during the session (as observed by several participants in the VR training group).

However, as with AR and traditional clinical training, VR presents some challenges. For example, a randomized trial involving the VR simulation of a mastoidectomy [29] showed that, in certain complex situations, VR clinical training results in reduced physical performance and mental agility in participants due to an inadequate ability to create a stereoscopic view of the clinical environment. This type of technological barrier is comparable to the situation in the present study, in which the participants reported some obstacles to their perception, such as the difficulty in handling the VR device during clinical training due to their never having used this technology previously, or simply because they found it uncomfortable. Indeed, albeit in unusual situations, studies have reported that participants may reject the experience due to unacceptable alterations (as when the device produces excessive warmth [30] or provokes cybernetic dizziness secondary to a lack of coordination between the visual, proprioceptive and vestibular information supplied [31], or because of heightened stress related to the individual’s safety while using the device [32], among other concerns).

Although studies related to the application of technological tools in clinical training have reported significant benefits for students [33,34,35], it has also been observed that these teaching approaches are rarely included in current study plans, since many training institutions have insufficient economic resources for this purpose [34]. Acknowledging this problem, one study [35] recommended that further research be conducted on this topic in order to highlight its importance, and another has discussed the economic impact of the creation of cost-free applications [22]. In short, the use of AR and VR is still a relatively new form of instruction in the field of urgent and emergency healthcare. From a pedagogical perspective, these technologies apply constructivism, with an active and contextual approach in which the student constructs their learning through experience, while behaviorism uses specific stimuli within the virtual environment to reinforce desired clinical behaviors through safe trial and error. From the cognitivist and connectivism perspectives, it helps organize mental schemas by linking them to prior knowledge and promotes internal and external collaborative learning networks [36].

A limitation of the present research was the impossibility of masking the intervention for the participants and for the researchers applying the intervention. Furthermore, the AR intervention was applied with a mobile device instead of purpose-built AR glasses due to the high cost of the latter. Also, the results should be considered with caution due to the limited validity and reliability of an ad hoc questionnaire for satisfaction.

## 5. Conclusions

In training for CPR and other forms of emergency and urgent healthcare, VR-based learning produced significantly better results in learning and user satisfaction than traditional methods and smartphone-based AR learning. No such significance was achieved by AR training compared with the control group, although the participants in the AR group scored more highly than those in the traditional training group. Pre-post differences in knowledge were significant in the AR and VR groups. The participants’ opinions about using VR and AR were good regarding motivation, ease of use and usefulness for learning.

Given the selection criteria applied and the randomization process followed, the results of this study could be generalized to a wider population of nurses working in emergency healthcare services and also to nursing students.

Regarding the applicability of the results obtained, educational centers that have the resources to obtain VR glasses should examine whether there exist VR applications compatible with their courses and which would improve the training provided. Further research in this field is needed to better determine the educational importance of VR and AR-based learning.

## Figures and Tables

**Table 1 healthcare-13-01034-t001:** Main characteristics of the study population (n = 93).

Study Variables		Group			*p*-Value
		AR (n = 31)	VR (n = 30)	Control (n = 32)	
Sex	Female	75.16%	75.16%	75.16%	0.846 Chi-square test
	Male	24.73%	24.73%	24.73%	
Age (years)	Maximum	30	31	31	0.794 ANOVA test
	Mean	25.09	25.70	25.41	
	Minimum	20	21	20	
Professional level	Student	19.38%	19.98%	28.12%	0.836 Chi-square test
	Graduate and Master’s Degree	77.52%	76.59%	68.25%	
	Ph.D.	3.23%	3.33%	3.12%	
Marital status	Married/partnered	71%	62%	63%	0.174 Chi-square test
	Single	29%	38%	37%	

**Table 2 healthcare-13-01034-t002:** Post-knowledge and satisfaction mean differences between groups.

Variable/Group	Mean (SD)	*p*	ANOVA (F)	Eta Squared (95%CI)
**Knowledge**		0.019	4.158	0.085 (0.02–0.195)
Control	7.96 (1.09)			
Augmented reality	8.19 (0.94)			
Virtual reality	8.66 (0.84)			
**Training satisfaction ***		0.066	2.802	0.059 (0.00–0.159)
Control	4.01 (0.84)			
Augmented reality	4.22 (0.61)			
Virtual reality	4.43 (0.67)			

Note: SD = Standard deviation; *p*: ANOVA test * = satisfaction was measured with a 5-point Likert scale question: How satisfied are you with the training you received? 1: Very dissatisfied. 2: Dissatisfied. 3: Neutral. 4: Satisfied. 5: Very satisfied.

**Table 3 healthcare-13-01034-t003:** Participants’ opinions on the use of AR/VR during healthcare training (mean scores in response to the ad hoc research questionnaire).

Item	Group	
	AR	VR
I felt stressed while using AR/VR.	1.2	1.8
I felt anxious while using AR/VR.	1.1	1.6
My attention was concentrated while using AR/VR.	4.7	4.8
I felt I was in control of the AR/VR simulation.	4.4	4.6
I liked using the AR/VR simulation.	5	5
Using the AR/VR simulation encouraged me to learn.	4.8	5
I take AR/VR seriously.	5	5
The AR/VR simulation of the health emergency was realistic.	4.1	4.4
It was interesting to use AR/VR.	5	5
I would recommend using AR/VR in other courses.	5	5
The use of AR/VR was intuitive	4.5	4.7
I didn’t enjoy using AR/VR in my professional training.	1.3	1.2
It was easy for me to control the AR/VR.	4.3	4.1
Using AR/VR hindered my professional training.	1.1	1.4
Using AR/VR complemented my professional training.	5	5
Using AR/VR helped me empathise with patients.	NA	2
I felt physically and mentally secure while using AR/VR.	5	5
I felt secure being able to make mistakes during AR/VR simulation.	5	5
I deliberately made mistakes during AR/VR simulation in order to learn more.	1.4	1
I was able to change my approach/try again/rectify mistakes during AR/VR simulation.	5	5
Using AR/VR improved my practical skills.	5	5
Using AR/VR improved my theoretical knowledge.	5	5
I applied operating protocols correctly during the use of AR/VR.	5	5
I have learnt from my mistakes during the use of AR/VR.	5	5
I completed the full scheduled AR/VR experience.	5	5
I had no problem with time management while using AR/VR.	5	5
Using AR/VR helped me prioritise different clinical situations.	3.9	4
I believe AR/VR makes it easier to acquire theoretical knowledge.	5	5
I believe AR/VR makes it easier to acquire practical knowledge.	4.8	5
I think AR/VR should be used more in healthcare training.	5	5
I agree with the use of AR/VR in healthcare training.	5	5
I would use AR/VR again in another training course.	5	5
I believe AR/VR will be of crucial importance in healthcare training in the future.	4.7	4.9
I believe that using AR/VR in healthcare training improves patient care.	4	4.3
I believe that using AR/VR in healthcare training benefits decision making in clinical practice.	3.9	4.0

Note: AR = augmented reality. VR = virtual reality.

## Data Availability

The data presented in this study are available on request from the corresponding author due to ethical reasons.

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
