# Peer review of "Effectiveness of Virtual and Augmented Reality for Emergency Healthcare Training: A Randomized Controlled Trial"

_healthcare, 2025, doi:10.3390/healthcare13091034_

Round 1

Reviewer 1 Report

Comments and Suggestions for Authors

Dear authors,

The manuscript's topic is very current and significant, and it is about the challenging issue of applying advanced technologies in emergency healthcare training. This study attempted to contribute and provide empirical data on multiple aspects of using advanced technology in healthcare training.

However, minor improvements are necessary for the manuscript to provide new, valid empirical data that will complement the existing literature on applying advanced technology in healthcare education.

 I would like to make suggestions for improving the manuscript:

Abstract: Although the abstract is structured, some revisions are needed. Ensure consistency in the reported sample size. Some sentences, such as the one on line 22 about the applied statistical software, are unnecessary. Please correct it.

Introduction: The introduction is coherent, easy to follow, and has a funnel-like structure. Revisions are necessary for the following:

To expand the facts about the application of advanced technology in healthcare education/training, it is necessary to provide specific examples: applications of these technologies, with which healthcare professionals, at what level of education, from which areas of education/training, as well as models, or levels of evaluation of the application of these technologies. Also, in addition to data on the application of advanced technologies in healthcare education/training at the global level, it is necessary to provide facts about their application in the region/country and institution where the study was conducted.

Materials and Methods

The methodology partially allows for the replication of the study.

Provide data on whether the created protocol meets the standards for reporting RCT trials (CONSORT).

A detailed description of the "intervention" procedure is adequate and essential. However, in the study variables and data collection part of the manuscript, a more detailed description of the instruments used to evaluate the variables in the study is necessary. Especially for the application of AR/VR and the knowledge test, which were theoretical frameworks (supported by references), how many questions the test contained, the criteria for scoring and ranking the results, whether pilot testing of the instrument was conducted, and whether and how the reliability of the instruments was confirmed. Please include this data.

Results

The results are generally adequately presented.

However, corrections are necessary for the tabular presentation of the data. In Table 1, it is necessary to add a column in which the statistical analysis data, which was the basis for calculating the p-value, will be listed. In Table 2, transfer the data in the table's header to the methodology/description of the instrument part of the manuscript and list the item instead. It is also necessary to add a column with ANOVA values and effect size data. Please include this data.

Discussion

The discussion is extensive, and the authors draw attention to many important questions raised by their research and the results of other authors, through a logical discussion structure with adequate references from the literature.

Conclusions

The conclusions are concise, well-argued, and based on research results.

References

The references mentioned are relevant to the topic that the paper dealt with.

I hope you find my comments helpful.

Author Response

Dear reviewer,

Thank you for your time for reviewing the paper and for your comments for its improvement. Please find below in “bold letter” the response to each comment. The changes in the manuscript have been highlighted in yellow

Dear authors,

The manuscript's topic is very current and significant, and it is about the challenging issue of applying advanced technologies in emergency healthcare training. This study attempted to contribute and provide empirical data on multiple aspects of using advanced technology in healthcare training.

However, minor improvements are necessary for the manuscript to provide new, valid empirical data that will complement the existing literature on applying advanced technology in healthcare education.

 I would like to make suggestions for improving the manuscript:

Abstract: Although the abstract is structured, some revisions are needed. Ensure consistency in the reported sample size. Some sentences, such as the one on line 22 about the applied statistical software, are unnecessary. Please correct it.

We have corrected the sample sizes and deleted the sentence following your recommendation.

Introduction: The introduction is coherent, easy to follow, and has a funnel-like structure. Revisions are necessary for the following:

To expand the facts about the application of advanced technology in healthcare education/training, it is necessary to provide specific examples: applications of these technologies, with which healthcare professionals, at what level of education, from which areas of education/training, as well as models, or levels of evaluation of the application of these technologies. Also, in addition to data on the application of advanced technologies in healthcare education/training at the global level, it is necessary to provide facts about their application in the region/country and institution where the study was conducted.

We have included more specifical information in the introduction as recommended.

Materials and Methods

The methodology partially allows for the replication of the study.

Provide data on whether the created protocol meets the standards for reporting RCT trials (CONSORT).

A detailed description of the "intervention" procedure is adequate and essential. However, in the study variables and data collection part of the manuscript, a more detailed description of the instruments used to evaluate the variables in the study is necessary. Especially for the application of AR/VR and the knowledge test, which were theoretical frameworks (supported by references), how many questions the test contained, the criteria for scoring and ranking the results, whether pilot testing of the instrument was conducted, and whether and how the reliability of the instruments was confirmed. Please include this data.

We have included more information in the methods about the questionnaires as recommended.

Results

The results are generally adequately presented.

However, corrections are necessary for the tabular presentation of the data. In Table 1, it is necessary to add a column in which the statistical analysis data, which was the basis for calculating the p-value, will be listed. In Table 2, transfer the data in the table's header to the methodology/description of the instrument part of the manuscript and list the item instead. It is also necessary to add a column with ANOVA values and effect size data. Please include this data.

We have deledted modified some “,” using “.”. We have deleted the space between some numbers and “%” and included the mentioned data following your recommendations.

Discussion

The discussion is extensive, and the authors draw attention to many important questions raised by their research and the results of other authors, through a logical discussion structure with adequate references from the literature.

Thanks for your comment. We have included some few sentences as recommended by other reviewers.

Conclusions

The conclusions are concise, well-argued, and based on research results.

Thanks for your comment.

References

The references mentioned are relevant to the topic that the paper dealt with.

 Thanks for your comment.

I hope you find my comments helpful.

Thanks a lot for your help and for your comments.

Reviewer 2 Report

Comments and Suggestions for Authors

The authors present a valuable study on the "Effectiveness of Virtual and Augmented Reality in Emergency Healthcare Training: A Randomized Controlled Trial."

In order to improve the quality of the manuscript, the authors should respond to the following reviewer comments and suggestions:

1. The "motivation" of this study should be made clear in the abstract and introduction. Only at the end do the authors report anything related to "this study is part of the first author's doctoral thesis."

2. If it is part of a doctoral thesis: What is the hypothesis of this study? The results are based on inductive logic, but do not pose new challenges that need to be resolved.

3. Sections of the introduction refer to several references, e.g.:
"In recent years, there has been an increase in research interest in see-through optical technologies, including products such as Microsoft HoloLens and Google Glass [5,7,9–11]." Are there 5 references for a line of analysis? The authors should better explain their background.

4. What are the new forms of training? The authors should include a section on pedagogy and didactics related to virtual reality and interaction with teaching.

5. The effectiveness metric responds to the need to reach a goal. What is that goal?

6. It is not clear how satisfaction was measured.

Author Response

Dear reviewer,

Thank you for your time for reviewing the paper and for your comments for its improvement. Please find below in “bold letter” the response to each comment. The changes in the manuscript have been highlighted in yellow

The authors present a valuable study on the "Effectiveness of Virtual and Augmented Reality in Emergency Healthcare Training: A Randomized Controlled Trial."

In order to improve the quality of the manuscript, the authors should respond to the following reviewer comments and suggestions:

  1. The "motivation" of this study should be made clear in the abstract and introduction. Only at the end do the authors report anything related to "this study is part of the first author's doctoral thesis."

We have included information at the end of the introduction. We did not have more words to add in the abstract.

  1. If it is part of a doctoral thesis: What is the hypothesis of this study? The results are based on inductive logic, but do not pose new challenges that need to be resolved.

We have included the hypothesis in the methods section.

  1. Sections of the introduction refer to several references, e.g.:
    "In recent years, there has been an increase in research interest in see-through optical technologies, including products such as Microsoft HoloLens and Google Glass [5,7,9–11]." Are there 5 references for a line of analysis? The authors should better explain their background.

We wanted to show that those studies were using those technologies. We have deteled some references in that sentence following your recommendation.

  1. What are the new forms of training? The authors should include a section on pedagogy and didactics related to virtual reality and interaction with teaching.

We have included information about it the discussion.

  1. The effectiveness metric responds to the need to reach a goal. What is that goal?

The goal is knowledge, the main dependent variable of the study.

  1. It is not clear how satisfaction was measured.

We have included more information in the methods sections about the satisfaction likert scale.

Reviewer 3 Report

Comments and Suggestions for Authors

This study is a randomized controlled trial that comparatively evaluates the effects of augmented reality (AR) and virtual reality (VR) technologies on emergency health services education. The originality of the study is that it simultaneously evaluates the effects of VR and AR on both knowledge level and participant satisfaction and does this with real application scenarios. It has the potential to contribute to the literature in terms of the integration of developing technologies into health education. However, there are some technical deficiencies in the methodological details of the study and the necessary corrections are listed below:
-Attention should be paid to punctuation marks. For example, Line 37.
-The contributions of this study to the literature should be listed at the end of the introduction section.
-Both nursing students and active nurses were included in the same group in the study. Considering that these two groups may show significant differences in terms of knowledge level and clinical experience, it would be useful to consider the effect of this situation in the analyses.
-It is not understood why some subheadings are given in square brackets. For example, Lines 132, 152. These should be corrected.
-The subjects of the training (MI, asthma, CPR) are specified, but it is not clear which knowledge levels are measured (theoretical or practical?).

-The content of the knowledge test used (how many questions it consists of, whether it is valid/reliable) should be specified in detail.

-The psychometric properties (validity, reliability) of the ad hoc scale used in the participant satisfaction measurement are not given. This reduces the reliability of the results.

-In Table 1, there is a space after the point in some percentages. In addition, the decimal separator "," should be replaced with ".". It is not appropriate to use both "." and "," in the study.

Author Response

Dear reviewer,

Thank you for your time for reviewing the paper and for your comments for its improvement. Please find below in “bold letter” the response to each comment. The changes in the manuscript have been highlighted in yellow.

This study is a randomized controlled trial that comparatively evaluates the effects of augmented reality (AR) and virtual reality (VR) technologies on emergency health services education. The originality of the study is that it simultaneously evaluates the effects of VR and AR on both knowledge level and participant satisfaction and does this with real application scenarios. It has the potential to contribute to the literature in terms of the integration of developing technologies into health education. However, there are some technical deficiencies in the methodological details of the study and the necessary corrections are listed below:
-Attention should be paid to punctuation marks. For example, Line 37.

We have reviewed all the punctuation marks following your recommendation.

-The contributions of this study to the literature should be listed at the end of the introduction section.

We have included information about it as indicated.

-Both nursing students and active nurses were included in the same group in the study. Considering that these two groups may show significant differences in terms of knowledge level and clinical experience, it would be useful to consider the effect of this situation in the analyses.

Due to the lower number of students and because there were no significant differences between groups at the baseline we did not do it.

-It is not understood why some subheadings are given in square brackets. For example, Lines 132, 152. These should be corrected.

We have corrected using the same subheadings all the time.

-The subjects of the training (MI, asthma, CPR) are specified, but it is not clear which knowledge levels are measured (theoretical or practical?). The content of the knowledge test used (how many questions it consists of, whether it is valid/reliable) should be specified in detail.

We have included more information about the knowledge questionnaire in the methods section.

-The psychometric properties (validity, reliability) of the ad hoc scale used in the participant satisfaction measurement are not given. This reduces the reliability of the results.

We have included as a limitation of the study.

-In Table 1, there is a space after the point in some percentages. In addition, the decimal separator "," should be replaced with ".". It is not appropriate to use both "." and "," in the study.

We have modified all.

Thanks a lot. Kind regards

Reviewer 4 Report

Comments and Suggestions for Authors

Thank you for the opportunity to review the paper titled " Effectiveness of virtual and augmented reality for emergency healthcare training: a randomized controlled trial. I believe this article would be of interest to readers as it covers an important and timely topic related to AI and healthcare. However, I believe some minor revisions are needed to improve the paper. I have listed my suggestions below.

Introduction

Line 53 and line 72, when you first introduce the AR and VR, it would be better to explain more clearly what the difference is between these two technologies.

Method

Line 118

“The final sample, with subjects that did not went to the training was n=93” , this statement is little confused, please state how many participants in your final sample.

Line 149

Can you clarify how Intervention group No. 2 (AR group) receive the training using iPhone 15 Pro Max? Are there the same training materials as intervention group 1 with different devices?

Line 155

Please explain “An ad hoc questionnaire was conducted” what is the hoc questionnaire, how many questions in it? Did it include the two open questions?

Line 161

Please clarify how to measure post-training knowledge, is it a questionnaire? What is the title of the questionnaire and what is its validity and reliability?

Please clarify how to measure satisfaction, is it a questionnaire? What is the title of the questionnaire and what is its validity and reliability?

Line 205

A proper pre-post test should use the same instrument to allow accurate comparison over time. Using different ranges or formats would make it difficult to assess change reliably. Please see the following statement:

“The initial questionnaire:  mean score for the AR group 4.16 points (range 2-7 points); VR group 4.20 points (range 2-8 points); control group 3.94 points (range 2-7 points).”

“final questionnaire: AR group 8.19 points (range 6-10 points); VR group 8.66 points (range 210 6-10 points); control group 7.98 points (range 6-10 points)?

Line 226

“Table 3 presents the mean satisfaction scores”

It is a little confusing how to measure the satisfaction scores? Which questionnaire measures the satisfaction scores?

Table 3 may not need to list all responses. Can it be summarized with category?

Author Response

Dear reviewer,

Thank you for your time for reviewing the paper and for your comments for its improvement. Please find below in “bold letter” the response to each comment. The changes in the manuscript have been highlighted in yellow.

Thank you for the opportunity to review the paper titled " Effectiveness of virtual and augmented reality for emergency healthcare training: a randomized controlled trial. I believe this article would be of interest to readers as it covers an important and timely topic related to AI and healthcare. However, I believe some minor revisions are needed to improve the paper. I have listed my suggestions below.

Introduction

Line 53 and line 72, when you first introduce the AR and VR, it would be better to explain more clearly what the difference is between these two technologies.

We have included more information about it in lines 77-79.

Method

Line 118

“The final sample, with subjects that did not went to the training was n=93” , this statement is little confused, please state how many participants in your final sample.

We have clarified the sentence.

Line 149

Can you clarify how Intervention group No. 2 (AR group) receive the training using iPhone 15 Pro Max? Are there the same training materials as intervention group 1 with different devices?

We have included a sentence to clarify it, the app were the same but one in group 1 in VR (with VR glasses) and in Group 2 in AR (With iphone)

Line 155

Please explain “An ad hoc questionnaire was conducted” what is the hoc questionnaire, how many questions in it? Did it include the two open questions?

We have included the requested information

Line 161

Please clarify how to measure post-training knowledge, is it a questionnaire? What is the title of the questionnaire and what is its validity and reliability?

Please clarify how to measure satisfaction, is it a questionnaire? What is the title of the questionnaire and what is its validity and reliability?

We have included more information about it as requested.

Line 205

A proper pre-post test should use the same instrument to allow accurate comparison over time. Using different ranges or formats would make it difficult to assess change reliably. Please see the following statement:

“The initial questionnaire:  mean score for the AR group 4.16 points (range 2-7 points); VR group 4.20 points (range 2-8 points); control group 3.94 points (range 2-7 points).”

“final questionnaire: AR group 8.19 points (range 6-10 points); VR group 8.66 points (range 210 6-10 points); control group 7.98 points (range 6-10 points)?

It was a word selection mistake. We wanted to show the minimum and the maximum score in each group. We have modified it. "AR group 4.16 points (minimum 2 points and maximum 7 points); VR group 4.20 points (minimum 2 points and maximum 8 points); control group 3.94 points (minimum 2 points and maximum 7 points)."

Line 226

“Table 3 presents the mean satisfaction scores”

It is a little confusing how to measure the satisfaction scores? Which questionnaire measures the satisfaction scores?

We have included more information about the 5 point likert-scale questionnaire.

Table 3 may not need to list all responses. Can it be summarized with category?

It cannot be summarized. We included all the responses to show as much information as possible.

Thanks a lot

Kind regards